# Ultrafast one-chip optical receiver with functional metasurface

Go Soma ✉, Tomohiro Akazawa, Eisaku Kato, Kento Komatsu, Mitsuru Takenaka, Yoshiaki Nakano & Takuo Tanemura ✉

High-speed optical receivers are crucial in modern optical communication systems. While complex photonic integrated circuits (PICs) are widely employed to harness the full degrees of freedom (DOFs) of light for efficient data transmission, their waveguide nature inherently constrains two-dimensional spatial scaling to accommodate a large number of optical signals in parallel. Here we present a scalable optical receiver platform that fully exploits the high spatial parallelism and ultrabroad bandwidth of light, while leveraging all DOFs—intensity, phase, and polarization. Our solution integrates a thin metasurface, composed of silicon nanoposts, with ultrafast membrane photodetectors on a compact chip. The metasurface provides all the functionalities of conventional PICs for normal-incident spatially parallelized light, enabling high-speed detection of optical signals in various modulation formats, including simultaneous detection of 320-gigabit-per-second four-channel four-level pulse amplitude modulation (PAM4) signals and coherent detection of 240-gigabit-per-second 64-ary quadrature amplitude modulation (64QAM) signals.

Metasurfaces (MSs) are two-dimensional (2D) arrays of subwavelength nanostructures that can manipulate the intensity, phase, and polarization of transmitted light through ultrathin flat elements, offering a compact alternative to conventional bulky optical systems[1–6]. With judiciously designed MSs, various functional devices have been realized, including achromatic lenses[7–11], polarimeters[12–17], color routers[18–21], holograms[22–29], and augmented/virtual reality and display devices[30–35]. While these imaging, sensing, and display applications have demonstrated the remarkable potential of MSs over the past decade, modern high-speed optical communication systems represent another promising frontier for leveraging the rich capabilities of MSs.

For instance, coherent optical transmission systems have enabled long-haul, high-capacity data transport by utilizing the full degrees of freedom (DOFs) of light—intensity, phase, and polarization—to encode information. Additionally, the space-division multiplexing (SDM) scheme, which employs spatially parallelized channels within multi-core and multi-mode fibers (MCFs/MMFs), is envisioned as the promising next-generation technology to further scale transmission capacity[36–38]. With the evolution of these paradigms, optical transceivers have become increasingly complex. A coherent receiver (CR), for example, requires multiple high-speed photodetectors (PDs), a polarization beam splitter (PBS), and precisely phase-controlled optical hybrids to retrieve dual-polarization in-phase and quadrature (IQ) components of optical signals[39]. However, today, these transceivers are implemented on waveguide-based photonic integrated circuits (PICs)[40–44], which are not easily scalable to a 2D array to accommodate a large number of spatial channels efficiently[45]. Although some attempts have been reported, demonstrating discrete MS-based devices for optical communication[46–52], MS-enabled fully integrated high-speed optical transceivers have not been realized to the best of our knowledge.

Here, we present a high-speed and spatially scalable optical receiver operating at the 1550-nm telecommunication wavelength that employs a functional dielectric MS integrated with an ultrafast membrane indium-gallium-arsenide (InGaAs) PD array (PDA). A micrometer-thick MS, composed of silicon (Si) nanoposts, provides all the necessary operations on normally incident light, offering functionalities equivalent to conventional millimeter-scale waveguide-based PICs. Combined with

School of Engineering, The University of Tokyo, Tokyo, Japan. ✉e-mail: go.soma@tlab.t.u-tokyo.ac.jp; takuo.tanemura@tlab.t.u-tokyo.ac.jp

the membrane InGaAs PD that can efficiently detect infrared light within a submicrometer thickness, we realize various types of ultrafast receivers on a compact chip that can accept spatially parallelized, normally incident optical signals directly from a multi-channel fiber without using bulky optics. High-speed signals in various formats, such as 240-Gbit/s 64-ary quadrature amplitude modulation (64QAM) and 320-Gbit/s four-channel four-level pulse amplitude modulation (PAM4), are successfully demodulated with a bit error rate (BER) of less than $8.8 \times 10^{-3}$, well below the soft-decision forward error correction (SD-FEC) threshold.

## Results

### Device concept and fabrication

Figure 1a shows the concept of our one-chip receiver platform, which comprises membrane InGaAs PDA and functional Si MS layers

integrated on both sides of a 525-μm-thick fused silica (SiO$_2$) substrate. The input signal light is normally incident on the MS side, which consists of a densely located array of 1050-nm-high Si nanoposts, operating as meta-atoms. The light transmitted through the MS layer is received by the membrane PDA, which consists of vertical p-i-n diodes with a 500-nm-thick i-InGaAs absorption layer sandwiched by p/n-doped InGaAs and indium phosphide (InP) layers (see Supplementary Fig. 1a for the detailed profile).

The PDA and MS layers of our platform enable unique properties unattainable with conventional surface-normal receivers. First, the membrane InGaAs/InP p-i-n PD provides efficient opto-electric (O-E) conversion in the 1550-nm wavelength band with ultralow capacitance and high electrical conductivity, resulting in ultrahigh O-E bandwidth exceeding 100 GHz[53,54]. Second, the Si MS layer offers

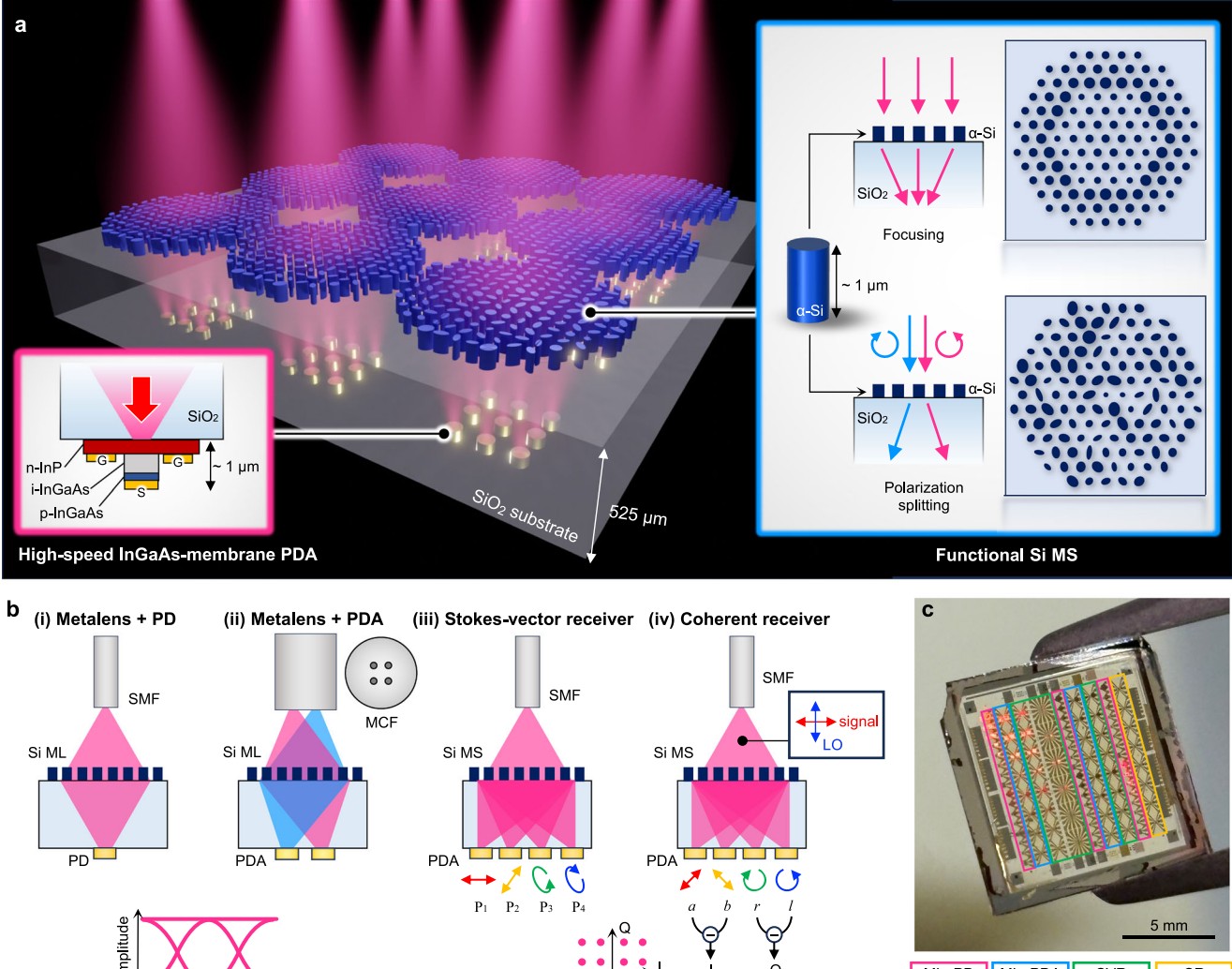

**Fig. 1 | One-chip optical receiver platform with an integrated Si metasurface and membrane InGaAs photodetector array. a** Schematic illustration of our receiver platform, where ultrathin amorphous-Si (α-Si) metasurface (MS) and high-speed membrane InGaAs photodetector (PD) layers are integrated on both sides of a transparent SiO$_2$ substrate. Spatially parallelized input signals are incident from the MS side and focused onto arrayed PDs. The MS offers various advanced functionalities, including focusing, splitting, and polarization manipulation, as shown in the right inset. The left inset shows the InGaAs/InP p-i-n structure of the high-speed membrane PD, which is directly bonded on the other side of the substrate. S signal, G ground. **b** Schematics of four types of receivers demonstrated in this work. (i) Single-channel metalens (ML)-integrated PD. (ii)

ML-integrated PD array (PDA) to detect parallel signals from a multi-core fiber (MCF). (iii) Stokes-vector receiver (SVR) with an integrated MS that sorts input light to four different polarization bases (P$_1$, P$_2$, P$_3$, and P$_4$) and focuses them on a four-channel PDA. (iv) Coherent receiver (CR) with an integrated MS that splits input light to four polarization states (a, b, r, and l) and focuses them on a four-channel PDA. The signal and local oscillator (LO) light from a single-mode fiber (SMF) are incident on the MS with x and y orthogonal polarizations, so that in-phase and quadrature (IQ) components of the signal can be retrieved from the four photocurrents. **c** Photograph of the receiver chip fabricated on a 1.2-cm-squared SiO$_2$ substrate, which contains 94 receivers with four different configurations.

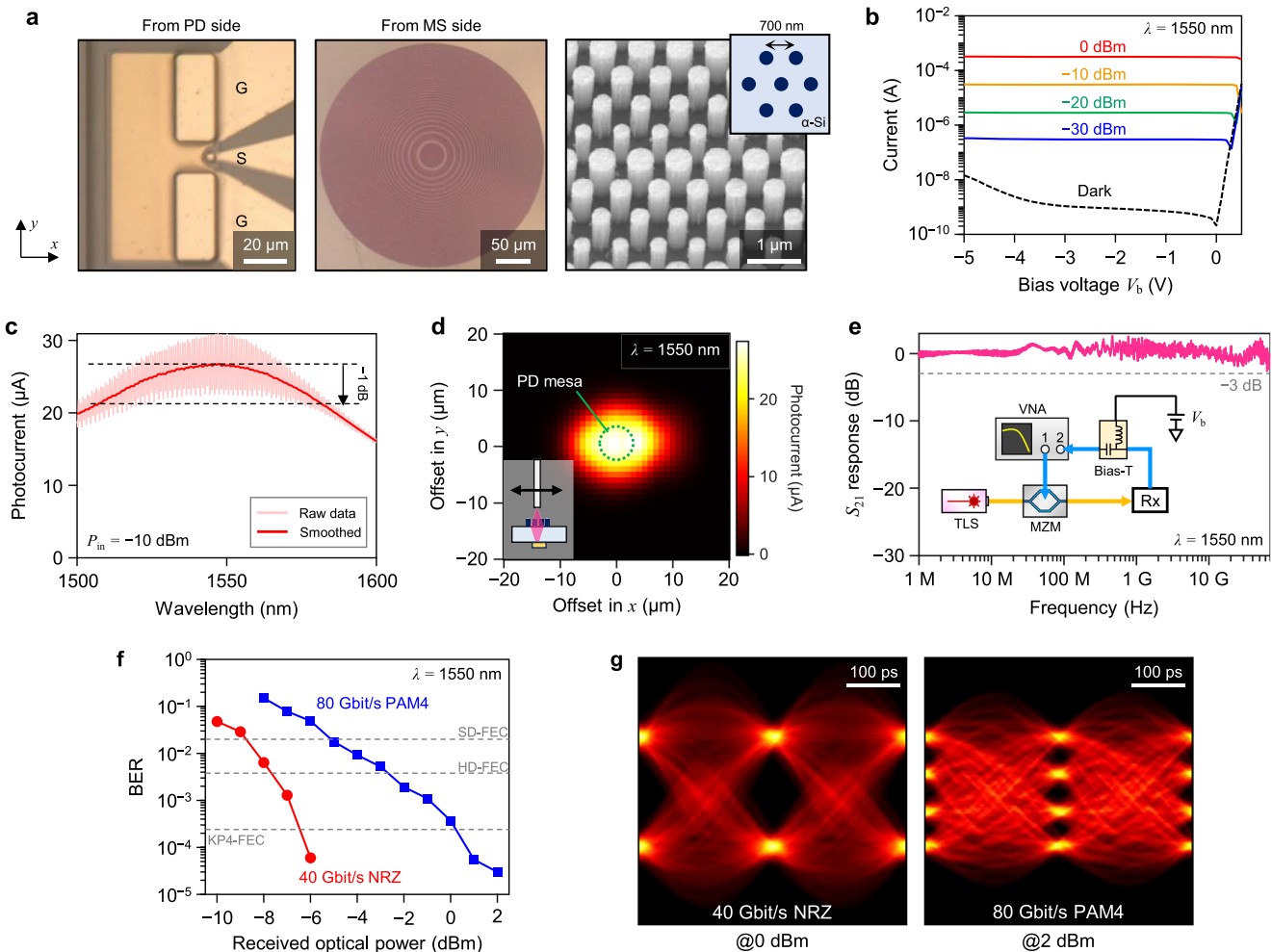

**Fig. 2 | Results of a metalens-integrated intensity-modulation and direct-detection receiver. a** Optical microscope and scanning electron microscope (SEM) images of the fabricated device, observed from the PD and MS sides of the chip. The diameter of the circular PD mesa is $D_{PD}$ = 6 μm. The MS comprises α-Si cylindrical nanoposts arranged on a triangular lattice with a lattice constant of 700 nm (inset). **b** Current-voltage (I–V) characteristics of the PD measured under various input optical power ($P_{in}$) at a wavelength of $λ$ = 1550 nm. **c** Photocurrent spectrum measured at $P_{in}$ = −10 dBm. The dark red line represents a smoothed curve of the raw data (bright red line). The oscillations are attributed to the Fabry–Pérot resonance between the output facet of the SMF and the $SiO_2$-InP interface at the membrane PD. **d** Dependence of receiver sensitivity on the in-plane position of the input SMF, measured at 1550-nm wavelength with $P_{in}$ = −10 dBm. The green dotted line shows the PD mesa with a diameter ($D_{PD}$) of 6 μm. **e** Frequency response measured at a bias voltage of $V_b$ = −3 V. The inset shows the schematic of the measurement setup. TLS tunable laser source, MZM Mach-Zehnder modulator, VNA vector network analyzer, Rx receiver. **f** Measured bit error rates (BERs) of 40-Gbit/s non-return-to-zero (NRZ) and 80-Gbit/s four-level pulse amplitude modulation (PAM4) signals at 1550-nm wavelength as a function of the received optical power. BER thresholds of various forward error correction (FEC) formats are also plotted as a reference. **g** Eye diagrams of received 40-Gbit/s NRZ and 80-Gbit/s PAM4 signals.

versatile functionalities beyond those of a simple focusing lens. By carefully designing the geometries of Si nanoposts, both the wavefront and polarization state of transmitted light can be controlled to enable advanced functionalities. These include beam splitting, polarization conversion, and demultiplexing, in addition to the basic functionality of a metalens (ML) to focus incident light onto a specific PD. We can, for example, design an MS to demultiplex multiple optical signals from an MCF and focus to different PDs (Fig. 1b-(ii)). Moreover, similar to PICs, complex optical components, such as a PBS and optical hybrid required in Stokes-vector receivers (SVRs) and CRs, can be realized in a surface-normal configuration (Fig. 1b-(iii), (iv)). The proposed platform, therefore, enables one-chip ultrafast optical receivers for normally incident SDM signals in various modulation formats without requiring additional optical components.

The one-chip receivers were fabricated by first transferring an InGaAs/InP membrane PD layer onto a $SiO_2$ substrate through the wafer-bonding-based technique that has been extensively developed for integrating III-V active devices with silicon photonics[55,56]. After the PD structures were fabricated, MS patterns were formed on an amorphous Si (α-Si) layer deposited on the other side of the substrate through electron-beam lithography and reactive-ion etching processes (detailed fabrication processes are provided in Methods and Supplementary Fig. 1b).

## Experimental results

Figure 1c shows the fabricated chip with a 1.2-cm-squared size, which contains 94 receivers in total with four different configurations: (i) single-channel ML-integrated intensity-modulation and direct-detection (IM-DD) receiver, (ii) four-channel ML-integrated IM-DD receivers to detect spatially parallelized signals from an MCF, (iii) SVR with a polarization-sorting MS, and (iv) CR with a polarization-splitting MS operating as an optical hybrid. The chip was mounted on a stage and characterized using radio-frequency (RF) probes (details of design and measurement for each device are provided in Methods and Supplementary Fig. 2).

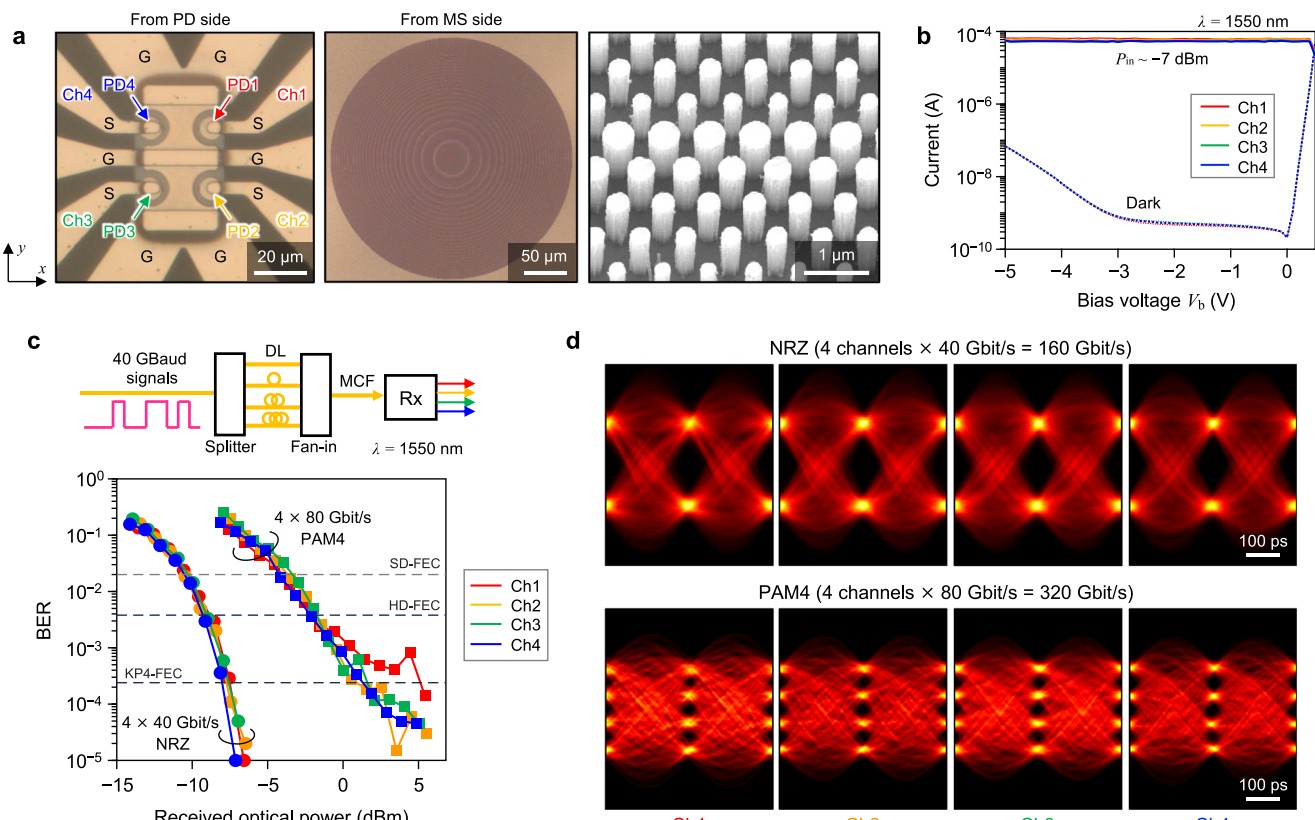

**Fig. 3 | Results of a metalens-integrated four-channel intensity-modulation and direct-detection receiver with a multi-core fiber. a** Optical microscope and SEM images of the fabricated device, observed from the PDA and MS sides of the chip. Ch channel. The diameter of each PD is $D_{PD} = 8$ μm. **b** Measured I–V curves of PDs when 1550-nm wavelength light with $P_{in} \sim -7$ dBm is incident from each core of an MCF. Dark currents of all PDs are also plotted. **c** Measured BERs of 40-Gbaud signals

for all channels at 1550-nm wavelength as a function of the received optical power. The top panel shows a schematic of the setup. A 40-Gbaud signal is split into four paths with different-length delay lines (DLs), transmitted through an MCF, and input to our receiver. **d** Eye diagrams of received 4 channels × 40-Gbit/s NRZ (160 Gbit/s) and 4 channels × 80-Gbit/s PAM4 (320 Gbit/s) signals.

**Metalens-integrated single-channel intensity-modulation and direct-detection receiver.** Figure 2 shows the first example of a single-channel IM-DD receiver, where a 340-μm-diameter MS, designed to work as an ML, is integrated on the other side of a 6-μm-diameter InGaAs/InP membrane PD. The MS is composed of $\alpha$-Si nanoposts arranged on a triangular lattice with a lattice constant of 700 nm (Fig. 2a, inset). They are designed to focus a beam emitted from a single-mode fiber (SMF) onto the PD. The distance between the output facet of the SMF and the MS is set to 841 μm, so the mode field diameter (MFD) of the beam incident to the MS is around 160 μm.

We first calculated the optical phase shift $\varphi_{ML}(x, y)$ required at each position $(x, y)$ on the MS to achieve the desired focusing functionality. The diameter of each cylindrical $\alpha$-Si nanopost was then determined to obtain $\varphi_{ML}(x, y)$ with a minimal insertion loss (see Methods and Supplementary Fig. 3). Rigorous full-wave simulation based on a finite-difference time-domain (FDTD) method shows that the designed ML exhibits the desired focusing function with a high focusing efficiency of 85% (see Supplementary Note 1 and Supplementary Fig. 4). Figure 2a shows optical microscope and scanning electron microscope (SEM) images of the fabricated device, observed from the MS and PD sides of the chip. Ground-signal-ground (GSG) electrode pads are integrated with the PD to enable high-speed characterization.

The current-voltage (I–V) characteristic is measured under different values of input optical power, $P_{in}$, at a wavelength of 1550 nm (Fig. 2b). The dark current is around 1 nA at a bias voltage of $V_b = -3$ V. From these results, the overall receiver sensitivity, which includes the focusing efficiency of ML, is derived to be 0.27 A/W. Using the

simulated absorption efficiency of around 27% by the membrane PD, the focusing efficiency of the fabricated ML is estimated to be around 80%. From the measured photocurrent spectrum (Fig. 2c), we can confirm that the device operates over a wide wavelength range with 1-dB and 3-dB bandwidths of 73 nm (1508–1581 nm) and over 100 nm, respectively. To examine the focusing functionality of the ML, the measured photocurrent is plotted as a function of the in-plane position of the input SMF in Fig. 2d. From this plot, it is confirmed that the sensitivity drops rapidly as we move the SMF by ±10 μm in both $x$ and $y$ directions. Since this sensitivity distribution should represent the convolutional integral of the incident beam profile at the PD plane and the aperture of the PD, having a mesa diameter of 6 μm, we can conclude that the fabricated MS effectively focuses the input beam with an MFD of around 160 μm down to around 15 μm at the input of the membrane PD.

Figure 2e shows the O-E frequency response of the device, measured by a vector network analyzer (VNA). The 3-dB bandwidth is more than 70 GHz, which was limited by the maximum frequency range of our VNA. Finally, Fig. 2f, g shows the results of the high-speed signal detection experiment. Although the fabricated device exhibited an ultrabroad bandwidth exceeding 70 GHz, we could only employ 40-Gbaud signals due to the bandwidth constraint of the real-time oscilloscope available at the time of the experiment. For both 40-Gbit/s non-return-to-zero (NRZ) and 80-Gbit/s PAM4 signals, BERs below the KP4-FEC threshold of $2.4 \times 10^{-4}$ are obtained with clear eye openings. In addition, wavelength-insensitive operation across the entire C band (1530–1570 nm) is confirmed (Supplementary Fig. 5).

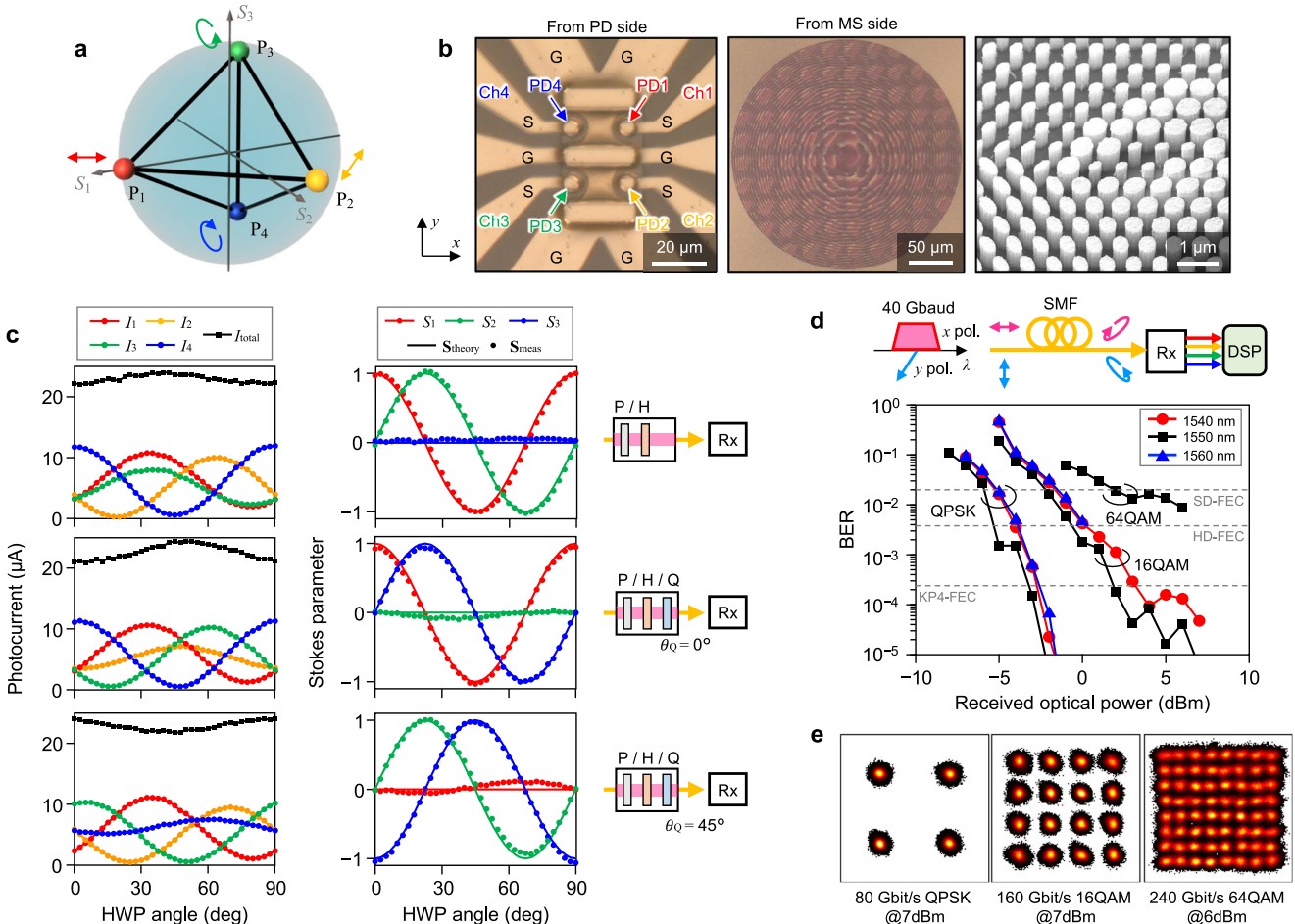

**Fig. 4 | Stokes-vector receiver with an integrated polarization-sorting meta-surface. a** Four polarization bases (P₁, P₂, P₃, P₄), used to project the input signal. They are set to constitute a regular tetrahedron on the Poincaré sphere so that the receiver sensitivity is maximized. **b** Optical microscope and SEM images of the fabricated device, observed from the PDA and MS sides of the chip. The diameter of each PD is $D_{PD}$ = 8 µm. **c** Measured photocurrents at four PDs $\mathbf{I} = (I_1, I_2, I_3, I_4)^t$ and their sum $I_{total} = \sum_{i=1}^{4} I_i$ (left panel) and retrieved Stokes parameters $\mathbf{S}_{meas} = (S_1, S_2, S_3)^t$ (center panel) as a function of the angle of a half-wave plate (HWP). The HWP is used to rotate the input polarization state in different configurations (right panel). The input power is $P_{in} \sim -10$ dBm. Theoretical curves ($\mathbf{S}_{theory}$)

are plotted by the solid lines. P: polarizer. H: half-wave plate. Q: quarter-wave plate. $\theta_Q$: quarter-wave plate angle. **d** Measured BERs of 40-Gbaud self-coherent signals as a function of the received optical power. Results at three different wavelengths of 1540, 1550, and 1560 nm are plotted for quadrature phase-shift keying (QPSK), 16-ary quadrature amplitude modulation (16QAM), and 64QAM formats. The top panel shows a schematic of the setup. A 40-Gbaud self-coherent signal is transmitted through an SMF and detected with a random polarization state by our SVR. **e** Constellation diagrams of received 80-Gbit/s QPSK, 160-Gbit/s 16QAM, and 240-Gbit/s 64QAM signals at 1550-nm wavelength.

**Metalens-integrated four-channel intensity-modulation and direct-detection receiver.** Figure 3 presents the results of extending our single-channel IM-DD receiver to four channels without additional components by exploiting the 2D spatial scalability of the surface-normal configuration. The fabricated device is shown in Fig. 3a. Here, four independent signals emitted from a four-core MCF are input to the device. The core pitch and MFD of the MCF are 40 µm and 10.4 µm, respectively. The separation between the MCF and MS is set to 631 µm, so the MFD of each channel incident on the MS is around 120 µm and overlaps with other channels. By designing the MS to operate as an ML with a focal length of 222 µm, the near-field image at the MCF facet is demagnified by ~0.54 and projected onto the PD plane. We can, therefore, detect four signals simultaneously using four-channel PDs with 8-µm circular mesas placed at 22-µm pitches (Fig. 3a).

Figure 3b shows the measured current-voltage (I–V) curves with and without light irradiation at a wavelength of 1550 nm. We confirm that identical characteristics are obtained for all PDs, and crosstalk from other channels is suppressed below −25 dB (see Supplementary Note 2 and Supplementary Fig. 6). Figure 3c, d shows the results of simultaneously receiving four-channel signals transmitted through the

MCF. Note that four signals are uncorrelated using optical delay lines (DLs) of different lengths, as shown in the top panel of Fig. 3c. BERs below the KP4-FEC threshold and clear eye diagrams are obtained for all four channels with 40-Gbit/s NRZ and 80-Gbit/s PAM4 formats, corresponding to 160-Gbit/s and 320-Gbit/s total data rates, respectively.

**Stokes-vector receiver with integrated polarization-sorting meta-surface.** Figure 4 shows the results of characterizing an SVR, which is capable of retrieving the polarization state of input light at high speed. To enable polarization sorting functionality by an MS, we employed elliptical Si nanoposts (Fig. 1a, right inset), arranged on a triangular lattice with a lattice constant of 700 nm. As the light emitted from an SMF transmits through the MS, it is decomposed into four different polarization bases, P₁, P₂, P₃, and P₄, and focused to respective PDs (Fig. 1b(iii)). From the four photocurrent signals, we can derive the full Stokes parameters of the input light through simple digital signal processing (DSP)[17].

It is proven that the maximum receiver sensitivity is obtained when the four polarization bases form a regular tetrahedron inscribed

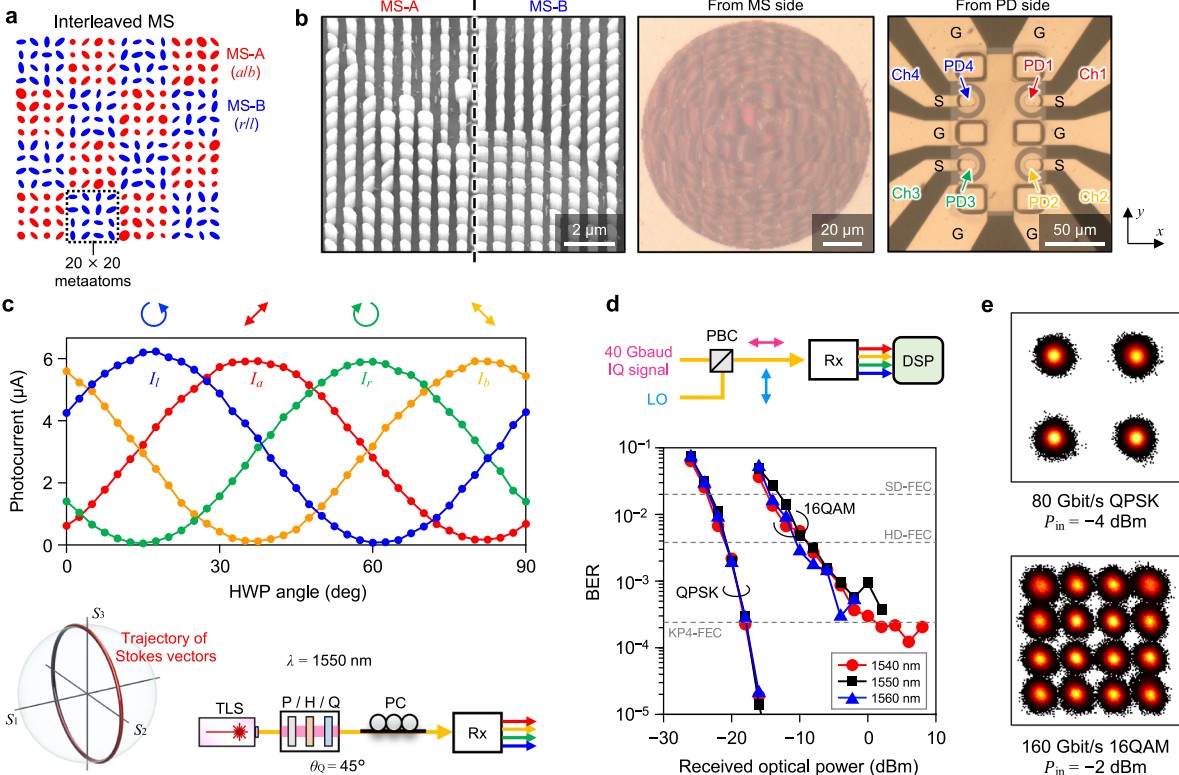

**Fig. 5 | Coherent receiver with an integrated metasurface operating as an optical hybrid. a** Schematic of the MS, where two independently designed sections, MS-A and MS-B, are interleaved. MS-A and MS-B split input light at ±45° linear polarization basis (*a/b*) and right- and left-handed circular polarization basis (*r/l*), respectively. **b** Optical microscope and SEM images of the fabricated device, observed from the MS and PDA sides. The diameter of each PD is $D_{PD} = 20 \mu m$. **c** Measured photocurrents $\mathbf{I} = (I_a, I_b, I_r, I_l)^t$ as a function of the HWP angle with

$P_{in} \sim -10$ dBm. The bottom panel shows the measurement setup and the trajectory of the input polarization state as the HWP is rotated. PC polarization controller. **d** Measured BERs of 80-Gbit/s QPSK and 160-Gbit/s 16QAM signals as a function of the received optical power at wavelengths of 1540, 1550, and 1560 nm. The top panel shows the setup. **e** Constellation diagrams of received 80-Gbit/s QPSK and 160-Gbit/s 16QAM signals at 1550-nm wavelength.

in the Poincaré sphere, as shown in Fig. 4a[12,17,57,58]. We therefore optimized the shape and rotation angle of elliptical Si nanoposts to achieve the Jones matrix distribution $\tilde{\mathbf{J}}(x, y)$ that provides such functionality[17] (see Methods and Supplementary Fig. 7 for detailed design methods). Full-wave simulation with the designed MS shows the desired focusing and polarization-sorting functions with a total focusing efficiency of 71% (see Supplementary Note 1 and Supplementary Fig. 8). Figure 4b shows the fabricated device.

Figure 4c shows the measured photocurrent at four PDs, $\mathbf{I} = (I_1, I_2, I_3, I_4)^t$ (left panel), and the Stokes vector, $\mathbf{S}_{meas} = (S_1, S_2, S_3)^t$ (center panel), which is retrieved from $\mathbf{I}$. The results are plotted as we rotate a half-wave plate (HWP) in three different configurations (right panel), so that the polarization state of input light is changed over the entire three-dimensional (3D) Stokes space. The actual Stokes vector, $\mathbf{S}_{theory}$, is also plotted with solid lines as a reference. The average error $\langle |\mathbf{S}_{meas} - \mathbf{S}_{theory}| \rangle$ is as small as 0.071.

Since our SVR operates at high speed, it can be employed to demonstrate self-coherent signal transmission by sending a high-speed coherent signal on one polarization and a non-modulated continuous-wave (CW) tone on the orthogonal polarization (see Methods and Supplementary Fig. 2d for details). Although the polarization state evolves randomly during transmission through a non-polarization-maintaining SMF, such a polarization change can be removed through DSP by detecting the full Stokes vector using our SVR (Fig. 4d, top panel).

Figure 4d shows the measured BERs of 40-Gbaud self-coherent signals in quadrature phase shift keying (QPSK), 16-ary quadrature amplitude modulation (16QAM), and 64QAM formats, corresponding to 80-Gbit/s, 160-Gbit/s, and 240-Gbit/s data rates, respectively. The constellation diagrams of the demodulated IQ signals at 1550-nm

wavelength are shown in Fig. 4e. Successful demodulation of up to 240-Gbit/s 64QAM signal at 1550-nm wavelength is achieved with a BER of less than $8.8 \times 10^{-3}$, which is below the 20% SD-FEC threshold. Furthermore, a polarization-drift-resilient operation is experimentally confirmed by receiving 160-Gbit/s 16QAM signals for various input states of polarization (Supplementary Fig. 9).

**Coherent receiver with an integrated metasurface.** Finally, Fig. 5 shows the results of a CR with an integrated MS that functions as an optical hybrid. As shown in Fig. 5a, the MS in this device is constructed by interleaving two sections: MS-A and MS-B. Each section is comprised of $20 \times 20$ meta-atoms, arranged on a square lattice with a lattice constant of 700 nm. MS-A and MS-B are designed to split the input light at ±45° linear polarization bases (*a/b*) and the right- and left-handed circular polarization bases (*r/l*), respectively, and focus them to four separate PDs (see Methods and Supplementary Fig. 10 for a detailed MS design).

Here, a high-speed coherent signal in one polarization state is combined with CW local oscillator (LO) light in the orthogonal polarization state and input to the device (Fig. 1b(iv)). They are then split into four polarization components, *a*, *b*, *r*, and *l*, by transmitting through the MS and detected by the four PDs with relative phases of 0°, 180°, 90°, and 270° between the signal and LO lightwaves, respectively. Based on a similar principle as the polarizer-based configuration[59], these PDs act as two sets of balanced PDs; by taking the differential signals between the *a/b* and *r/l* PDs (i.e., $I_a - I_b$ and $I_r - I_l$), we can retrieve the in-phase and quadrature components of the input signal light, respectively. We should note that, unlike the configuration using arrayed polarizers[59], this scheme with a polarization-splitting MS does not suffer from a 3-dB intrinsic loss. While we employ an off-chip

polarization beam combiner (PBC) to combine the signal and LO in this work for the convenience of measurement, the MS could include the beam-combining functionality as well[50] to realize a single-chip device. The dimensions and rotation angles of the elliptical $\alpha$-Si nanoposts in MS-A and MS-B were designed to realize desired focusing functionalities for corresponding polarization bases[25,26]. Full-wave simulation with the designed MS shows a high focusing efficiency of 87% and low crosstalk below −20 dB (see Supplementary Note 1 and Supplementary Fig. 11). Figure 5b shows the fabricated device, which contains four membrane PDs separated by 53 μm and integrated with a 120-μm-diameter MS on the other side.

Figure 5c shows the results of characterizing the polarization-splitting functionality of the MS. The photocurrents at four PDs are plotted as functions of the HWP angle (see the bottom panel in Fig. 5c for the setup and trajectory of the input polarization state). Sinusoidal responses are obtained in agreement with the theory, demonstrating that both MS-A and MS-B sections operate properly. The extinction ratio of over 15 dB is obtained for all ports. The rather small total responsivity (~0.13 A/W), corresponding to the focusing efficiency of ~40%, is attributed to imperfect MS fabrication and alignment error of the input fiber. Finally, Fig. 5d, e shows the results of high-speed coherent detection experiments using our device. Successful demodulation with a BER well below the 7% hard-decision FEC (HD-FEC) threshold of $3.8 \times 10^{-3}$ is achieved for both 80-Gbit/s QPSK and 160-Gbit/s 16QAM signals at 1540-, 1550-, and 1560-nm wavelengths (Fig. 5d). Clear constellation diagrams of demodulated IQ signals are obtained for all cases (Fig. 5e).

## Discussion

We have demonstrated an optical receiver platform with high spatial parallelism, consisting of an ultrathin dielectric MS and high-speed PDA integrated on a compact chip. Unlike conventional PIC-based receivers, our surface-normal platform offers a highly scalable solution for detecting a massive number of coherent optical channels without additional components (see discussion on the comparison between these platforms in Supplementary Note 3). This was achieved through wafer bonding of an ultrafast (>70 GHz) membrane InGaAs/InP p-i-n PD layer to a SiO$_2$ substrate, along with backside integration of a functional MS composed of Si nanoposts. The MS performs all the essential operations, including beam splitting, polarization sorting, and focusing, on normally incident light. Using the fabricated chip, we have successfully demonstrated the simultaneous detection of 320-Gbit/s four-channel PAM4 signals from a four-core MCF without using a fan-out device. Additionally, we achieved demodulation of high-speed coherent signals in various formats, including 80-Gbit/s QPSK, 160-Gbit/s 16QAM, and 240-Gbit/s 64QAM.

While the integration of MSs with active optoelectronic devices has been demonstrated for low-speed applications, such as ML-integrated image sensors[60–62], mid-infrared photodiodes[63,64], and vertical-cavity surface-emitting lasers[65–68], this work is, to our knowledge, the first to show that the integration of a functional MS with an ultrafast PDA unlocks new possibilities of MS technologies for advanced optical communication transceivers by providing 2D spatial scalability within a compact chip. Moreover, our versatile platform holds great promise for a wide range of high-speed applications that leverage spatial parallelism of light, including ultra-dense optical interconnects[69–71], free-space optical communication[72–75], large-scale optical neural networks[76–78], and coherent 3D imaging[79–81].

## Methods

### Device fabrication

First, the epitaxial InGaAs/InP layers, consisting of p-i-n diode and sacrificial layers, were grown on a 2-inch InP substrate by metal-organic vapor phase epitaxy (MOVPE). The detailed layer structure is provided in Supplementary Fig. 1a. The device was then fabricated through the following steps, as shown in Supplementary Fig. 1b.

1. *Wafer bonding*: The 2-inch InP substrate with p-i-n epitaxial layers was bonded on a 3-inch fused silica (SiO$_2$) substrate with an Al$_2$O$_3$ bonding interface[56].

2. *Removal of InP substrate and sacrificial layers*: The InP substrate and the sacrificial InP/InGaAs layers were removed through wet chemical etching using HCl (for InP) and H$_3$PO$_4$ + H$_2$O$_2$ (for InGaAs) solutions. After this process, thin InGaAs/InP layers for the p-i-n membrane PDs remained on the SiO$_2$ substrate.

3. *$\alpha$-Si deposition*: A 1050-nm thick $\alpha$-Si layer was deposited by plasma-enhanced chemical vapor deposition (PECVD) on the backside of the SiO$_2$ substrate, which is then protected with a polyimide layer. The bonded wafer was diced into 1.2 cm squared chips for subsequent device processes.

4. *p-contact metal formation*: Ti/Au (30/150 nm) layers for contacting the p$^+$-InGaAs layer were formed by photolithography, electron-beam (EB) evaporation, and liftoff process, followed by annealing at 300 °C for 1 min to reduce the contact resistance.

5. *Mesa formation*: The PD mesas were formed through wet chemical etching with HCl + H$_3$PO$_4$ (for InP) and H$_2$SO$_4$ + H$_2$O$_2$ (for InGaAs). Then, the larger mesas for n-contact were formed by photolithography and wet chemical etching.

6. *n-contact metal formation*: Ni/Ti/Au (30/20/150 nm) layers for contacting the n$^+$-InGaAs layer were formed in the same manner as the p-contact metal formation and annealed at 300 °C for 1 min.

7. *Passivation*: After the native oxides on the surface were removed by buffered hydrofluoric acid (BHF) for 1 min, the surface was passivated through (NH$_4$)$_2$S$_x$ treatment for 30 min at room temperature[82]. Then, a photosensitive polyimide layer (LT-S8010A, Toray) was coated over the device. After forming the contact openings through photolithography, the device was cured at 230 °C for 1 hour under a nitrogen atmosphere.

8. *Electrode formation*: Ti/Au (50/450 nm) electrode patterns were formed by photolithography, EB evaporation, and liftoff processes.

9. *Protection layer formation*: A thick photoresist layer (SU-8 3005, Kayaku Advanced Materials) was spin-coated on the device as a protection layer. The contact openings at the electrode pads were formed through photolithography.

10. *Alignment mark formation*: The chip was flipped, and the backside was cleaned through the O$_2$ ashing process. The alignment marks were then formed through photolithography with backside alignment to the PD patterns, followed by reactive-ion etching (RIE) of the $\alpha$-Si layer.

11. *EB lithography*: The MS patterns were defined by EB lithography (F7000S, ADVANTEST) using negative EB resist (OBER-CAN038, Tokyo Ohka Kogyo) and developer (NMD-W).

12. *MS formation*: The MS patterns on EB resist were transferred to the $\alpha$-Si layer by RIE (MUC-21 ASE-SRE, Sumitomo Precision Products) with SF$_6$ and C$_4$F$_8$ gases, known as the Bosch process, followed by the O$_2$ ashing process. In future work, a capping layer can be employed on the MSs to enhance device stability and environmental robustness.

### Metasurface design

**Metalens.** To design an ML that focuses incident lightwave to a PD at a desired position, we first calculate the phase shift required at each position $(x, y)$ of the MS. It is described as the sum of two phase profiles: $\varphi_{\mathrm{ML}}(x, y) = \varphi_{\mathrm{col}}(x, y) + \varphi_{\mathrm{foc}}(x, y)$. The first term $\varphi_{\mathrm{col}}(x, y)$ corresponds to the function of collimating a spherical lightwave from a fiber core located at a distance $f_1$ and is expressed as

$$\varphi_{\mathrm{col}}(x, y) = -\frac{2\pi}{\lambda}\left(\sqrt{x^2 + y^2 + f_1^2} - f_1\right), \qquad (1)$$

where $\lambda$ is the wavelength. On the other hand, the second term $\varphi_{\text{foc}}(x, y)$ corresponds to the spherical phase profile to achieve the function of a focusing lens with a focal length of $f_2$ inside the SiO$_2$ substrate. Using $n_s$ to denote the refractive index of SiO$_2$, $\varphi_{\text{col}}(x, y)$ is expressed as

$$\varphi_{\text{foc}}(x, y) = -\frac{2n_s\pi}{\lambda}\left(\sqrt{x^2 + y^2 + f_2^2} - f_2\right). \quad (2)$$

In this work, we set $\lambda = 1550$ nm, $n_s = 1.53$, and $f_2 = 525$ μm, which is the thickness of the SiO$_2$ substrate. For each device, we adjusted $f_1$ so that the MFD at the PD plane $w_{\text{PD}}$ becomes $1/\sqrt{2}$ times the PD diameter $D_{\text{PD}}$. $w_{\text{PD}}$ is given as $w_{\text{PD}} = w_{\text{fiber}}(f_1/f_2)/n_s$, where $w_{\text{fiber}}(=10.4\,\mu\text{m})$ is the MFD of the input fiber. The phase profile $\varphi_{\text{ML}}(x, y)$ used to design the ML of the receiver in Fig. 2 is depicted in Supplementary Fig. 3a.

Next, to determine the geometrical shapes of the meta-atoms to achieve $\varphi_{\text{ML}}(x, y)$, we simulated the transmission characteristics of Si nanoposts using the rigorous coupled-wave analysis (RCWA) method[83]. Namely, we calculated the complex amplitude $\tilde{t}$ of light transmitted through a periodic array of 1050-nm-high circular Si nanoposts placed on a triangular lattice with a lattice constant of 700 nm, as shown in Supplementary Fig. 3b. Here, the refractive indices of SiO$_2$ and Si were set to 1.53 and 3.392, respectively. Simulated transmittance and phase properties ($|\tilde{t}|^2$ and $\angle\tilde{t}$) are plotted as a function of the meta-atom diameter $D$ in Supplementary Fig. 3c. We can confirm that an arbitrary phase change ($0 - 2\pi$) can be obtained with a sufficiently high transmittance ($>0.89$) by judiciously selecting $D$ from 204 to 499 nm. We could, therefore, derive the shape of the meta-atom at each position $(x, y)$ by mapping from $\varphi_{\text{ML}}(x, y)$ to $D(x, y)$ using Supplementary Fig. 3d. Supplementary Fig. 3e shows the distribution of the derived meta-atom diameters $D(x, y)$.

**Metasurface for Stokes-vector receiver.** The polarization-sorting MS used in our SVR was designed based on our previous work[17]. The Jones matrix of the meta-atom used in this work, which is non-chiral and ideally lossless, can be expressed in a general form as[25,26]

$$\tilde{\mathbf{J}}_{\text{MA}} = \mathbf{R}(\theta)\begin{pmatrix} e^{i\varphi_u} & 0 \\ 0 & e^{i\varphi_v} \end{pmatrix}\mathbf{R}(-\theta), \quad (3)$$

where $\varphi_u$ and $\varphi_v$ represent the phase shifts for the eigenmode polarized along the fast and slow axes of the meta-atom, respectively, and $\mathbf{R}(\theta)$ is a rotation matrix with an angle $\theta$ of the meta-atom. From Eq. (3), we can understand that an arbitrary symmetric unitary matrix can be realized by judiciously designing three meta-atom parameters ($\varphi_u, \varphi_v, \theta$).

We analytically derived $\tilde{\mathbf{J}}_{\text{MA}}$ of each meta-atom at $(x, y)$ to realize the polarization-sorting and focusing functionalities to four PDs (see ref. 17 for detailed derivations). The obtained distributions of three parameters, $\varphi_u(x, y)$, $\varphi_v(x, y)$, and $\theta(x, y)$, of $\tilde{\mathbf{J}}_{\text{MA}}$ are depicted in Supplementary Fig. 7a. To derive the actual dimensions ($D_u, D_v$) of each meta-atom from the optical parameters ($\varphi_u, \varphi_v$), we simulated the transmission of a periodic array of Si nanoposts for $x$- and $y$-polarization input at 1550-nm wavelength using rigorous coupled-wave analysis (RCWA)[83]. The simulated transmission coefficients $\tilde{t}_u(D_u, D_v)$ and $\tilde{t}_v(D_u, D_v)$ are shown in Supplementary Fig. 7c. Using these results, we derived the geometrical parameters ($D_u(\varphi_u, \varphi_v), D_v(\varphi_u, \varphi_v)$) to satisfy the target phase shifts ($\varphi_u, \varphi_v$) with a high transmittance (Supplementary Fig. 7d)[25,48]:

$$(D_u(\varphi_u, \varphi_v), D_v(\varphi_u, \varphi_v)) = \underset{D_u, D_v}{\text{argmin}}\left\{|\tilde{t}_u(D_u, D_v) - e^{i\varphi_u}|^2 + |\tilde{t}_v(D_u, D_v) - e^{i\varphi_v}|^2\right\}, \quad (4)$$

Here, $D_u$ and $D_v$ were limited from 200 to 600 nm for ease of fabrication. Supplementary Fig. 7e shows the distributions of the derived geometric parameters: $D_u(x, y)$ and $D_v(x, y)$.

**Metasurface for coherent receiver.** For the MS used in our CR, two interleaved sections, MS-A and MS-B, were designed independently. Here, MS-A and MS-B, comprised of $20 \times 20$ meta-atoms (see Fig. 5a), were designed to function as polarization splitters for $\pm45°$ linear ($a/b$) and circular ($r/l$) polarization bases, respectively. The phase profiles $\varphi_p(x, y)$ ($p = a, b, r, l$) required to focus the input lightwave from a fiber to the center position $(x_p, y_p)$ of the corresponding PD are expressed as

$$\varphi_p(x, y) = \varphi_{\text{col}}(x, y) - \frac{2n_s\pi}{\lambda}\left(\sqrt{(x - x_p)^2 + (y - y_p)^2 + f_2^2} - f_2\right). \quad (5)$$

For MS-A, the three parameters of each meta-atom were determined in a straightforward manner as $\varphi_u = \varphi_a$, $\varphi_v = \varphi_b$, and $\theta = \pi/4$. On the other hand, the parameters for MS-B were obtained by $\varphi_u = (\varphi_r + \varphi_l)/2$, $\varphi_v = (\varphi_r + \varphi_l)/2 + \pi$, $\theta = (\varphi_r - \varphi_l)/4$[13]. Supplementary Fig. 10a shows the distributions of the optical parameters ($\varphi_u, \varphi_v, \theta$). We then derived the shapes of all meta-atoms using the mapping tables, similar to the design procedure for SVR. In this case, we employed a periodic nanopost array on a square lattice with a lattice constant of 700 nm (see Supplementary Fig. 10b–d). The distributions of the derived geometrical parameters ($D_u(x, y), D_v(x, y)$) are shown in Supplementary Fig. 10e.

**Bandwidth measurement**
The setup is shown in the inset of Fig. 2e. The transmitter consisted of a tunable laser source (TLS) and a Mach-Zehnder modulator (MZM) (MX70G, Thorlabs), which was driven by an electrical signal from port 1 of a vector network analyzer (VNA) (MS4647B, Anritsu). The modulated signal at 1550 nm was received by our fabricated device, which was biased through a bias-T (BT45R, SHF). The alternating current-coupled signal from the membrane PD was sent to port 2 of the VNA.

**High-speed signal detection experiments**
**Intensity modulation and direct detection experiment.** The experimental setups are shown in Supplementary Fig. 2b, c. CW light from a TLS (TSL-510, Santec) was intensity-modulated by an MZM, which was driven by a Nyquist-filtered electrical signal from an arbitrary waveform generator (AWG) (M8196A, Keysight: 32 GHz, 92 GS/s). The modulated optical signal was amplified by an erbium-doped fiber amplifier (EDFA) and filtered by an optical bandpass filter (OBPF). The input optical power to the receiver was controlled by a variable optical attenuator (VOA). The membrane PDs on the fabricated devices were DC-biased at $V_b = -3$ V through bias-Ts. The electrical signals from the PDs were captured by a real-time oscilloscope (UXR0204A, Keysight: 20 GHz, 128 GS/s). We applied DSP-based equalization to the signals to compensate for the effect of the bandwidth limits of the measurement system, including those of the MZM and RF components. The number of filter taps was chosen to be five and nine for NRZ and PAM4 signal formats, respectively.

For the IM-DD experiment using an MCF (Supplementary Fig. 2c), the optical signal after the VOA was split into four channels by a fiber coupler. They were transmitted through fiber DLs with different lengths so that the signals in different MCF cores were uncorrelated. Four-channel photocurrent signals from the receiver were simultaneously captured by the real-time oscilloscope and demodulated through DSP.

**Self-coherent experiment using Stokes-vector receiver.** The setup is shown in Supplementary Fig. 2d. CW light from the laser (TSL-510, Santec) was split by a 50:50 fiber coupler into two paths to generate a self-coherent signal. The light in the signal path was modulated using a

lithium-niobate-based IQ modulator (Ftm7962ep, Fujitsu) driven by two-channel Nyquist-filtered signals from the AWG. The CW power in the other path was adjusted by a VOA to the same level as the signal power. They were then combined by a PBC, amplified by an EDFA, filtered by an OBPF, and input to the fabricated SVR. The input optical power was controlled by another VOA. The photocurrent signals from the membrane PDA were captured by the real-time oscilloscope. Through offline DSP with $2 \times 3$ real-valued multi-input-multi-output (MIMO) equalizers[84,85], polarization changes inside the fiber were removed, and the original IQ signals were retrieved.

To emulate arbitrary polarization drift during transmission, we employed the setup shown in Supplementary Fig. 9a. A polarization controller (PC), composed of an HWP and a quarter-wave plate (QWP), was inserted to change the polarization state of the self-coherent signal input to the device.

**Coherent detection experiment.** The experimental setup is shown in Supplementary Fig. 2e. CW light from a narrow-linewidth laser (TLG-210, Alnair Labs) was modulated by the IQ modulator, driven by two-channel Nyquist-filtered signals from the AWG. The modulated coherent optical signal was amplified by an EDFA, filtered by an OBPF, and its power was adjusted by a VOA. Then, an LO light from another laser was combined by a PBC and input to the fabricated device. Here, the incident signal and LO light were fixed to the $x$ and $y$ polarization states, respectively. The photocurrent signals from four PDs were captured by the real-time four-channel oscilloscope. Through offline DSP, the differences between the signals from $a/b$ and $r/l$ PDs were derived, followed by a $2 \times 2$ adaptive MIMO filter for equalization and retrieval of original IQ signals. Note that since the optical hybrid function is identical to that in the standard coherent receiver, we can apply the same DSP to retrieve the IQ signals.

## Data availability
All the data generated in this study are available via *Zenodo* at https://doi.org/10.5281/zenodo.17422837(ref. [86]).

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

## Acknowledgements

This work was obtained in part from the commissioned researches (No. JPJ012368C08801 (T.T.) and JPJ012368C03601 (T.T.)) by the National Institute of Information and Communications Technology (NICT), Japan, and was partially supported by Japan Society for the Promotion of Science (JSPS) KAKENHI, Grant Numbers JP23H05444 (T.T.), JP23H00172 (M.T.), JP23H00272 (Y.N.), and JP24KJ0557 (G.S.), and World-leading Innovative Graduate Study Program–Quantum Science and Technology Program (WINGS–QSTEP), the University of Tokyo (G.S.). Part of the device fabrication and characterization was conducted at the cleanroom facilities of d.lab at the University of Tokyo, supported by the Nanotechnology Platform Program of the Ministry of Education, Culture, Sports, Science and Technology (MEXT), Japan, Grant Numbers JPMXP1224UT1115 (T.T.) and JPMXP1223UT0179 (T.T.). The authors thank K. Konishi, K. Misumi, M. Hino, and H. Miyano for their support in device fabrication and characterization.

## Author contributions

G.S. and T.T. conceived the experiment. G.S. performed device design, fabrication, measurement, and data analysis. T.A. performed the wafer bonding process. E.K. performed the MOVPE crystal growth. K.K. developed the MS fabrication process. M.T. contributed to the discussion of the bonding process and epitaxial layer design and provided experimental facilities. Y.N. contributed to the overall discussion and provided experimental facilities. G.S. and T.T. wrote the manuscript with inputs from all authors. T.T. supervised the project.

## Competing interests

The authors declare no competing interests.
