## [Transparent Peer Review file · Nature Communications]

Ultrafast one-chip optical receiver with functional metasurface

Corresponding Author: Professor Takuo Tanemura

Version 0:

Reviewer comments:

Reviewer #1

(Remarks to the Author)

This manuscript presents a novel optical receiver platform integrating a functional dielectric metasurface with an ultrafast membrane photodetector array, enabling high-speed, parallel detection of multi-channel and multi-modulation-format optical signals. The work demonstrates significant advancements in addressing scalability limitations of traditional photonic integrated circuits and holds promise for applications in space-division multiplexing and coherent optical communications. Below are the areas for improvement.

1. The metasurface design relies heavily on empirical phase and polarization models without rigorous electromagnetic verification. Finite-difference time-domain (FDTD) or rigorous coupled-wave analysis (RCWA) simulations are absent, leaving uncertainties regarding phase uniformity, cross-polarization leakage, and near-field coupling effects. These omissions weaken confidence in the design's reproducibility and scalability.
 2. Critical data on environmental robustness—such as thermal cycling, humidity resistance, and long-term operation—are missing. Such information is vital for assessing practical viability, particularly in harsh telecom environments.
 3. The multi-channel receiver's performance metrics lack quantification of adjacent channel interference (e.g., adjacent channel isolation ratio, ICR). Without this data, it is challenging to evaluate the metasurface's effectiveness in suppressing inter-channel crosstalk, a key determinant of system-level bit-error-rate (BER).
 4. The manuscript fails to contextualize the proposed design against established PIC-based receivers (e.g., silicon photonic coherent engines). Direct comparisons of key metrics—power consumption, footprint, latency, or cost—would clarify the innovation's advantages and limitations.
 5. Experiments are confined to static laboratory conditions (fixed wavelength, polarization, and input power). Real-world applications demand resilience to dynamic perturbations (e.g., polarization drift, wavelength hopping, or fiber nonlinearity). Supplementary tests or discussions on digital signal processing (DSP)-enabled calibration would enhance practical relevance.
 6. Key figures (e.g., Fig. 2c) lack error bars/confidence intervals, raising concerns about data reliability. Additionally, incomplete definitions of acronyms (e.g., PD, MS) and sparse methodological details hinder reproducibility.
- This work pioneers a metasurface-integrated optical receiver platform with transformative potential for high-speed, parallel optical communications. However, the absence of rigorous theoretical validation, environmental stability data, and comparative benchmarks limits its impact. Addressing these gaps—through supplementary simulations, and expanded experimental comparisons—would elevate the manuscript to a publishable standard in Nature Communications.

Reviewer #2

(Remarks to the Author)

In this work, the authors present a proof of concept for several integrated metasurface enabled optical receivers for an assortment of compact signal multiplexing schemes. The authors design transmission-mode metasurfaces to image the output of optical fibers onto membrane photodetectors polarization-selectively. This manuscript showcases innovative techniques that advance a new avenue for fabricating high bandwidth optical receivers. I think that the manuscript is suitable for publication in Nature Communications provided some revisions are made.

1. The authors' claim is motivated by the alleged advantage of metasurfaces over conventional PICs, specifically in "their waveguide nature inherently constrain[ing] two dimensional spatial scaling to accommodate a large number of optical signals in parallel." The authors should provide an analysis of the on-chip footprint of equivalent photonic integrated circuitry

(injected by, say, an edge coupled fiber) for each device and its scaling behavior for high parallelism.

2. Data in Figs. 2c and 2e are plotted somehow with a variable thickness line, which at some points seems as if it is multivalued with respect to the frequency axis. The authors should verify that the data is valid and plot again with conventional data lines and data markers.

3. In Figure S3b, the transmitted phase is calculated as a function of the structural parameter D . However, Figure S5b presents the structural parameter as a function of desired $\phi_u(x, y)$ and $\phi_v(x, y)$ instead of vice versa. Additionally it is claimed that this mapping is “one-to-one” on line 422 and elsewhere. This is highly unlikely, seeing that the relation shown in S3b is not one-to-one. The authors should show ϕ_u , ϕ_v , and T as a function of D_u and D_v , and remove the language “one-to-one.”

4. On page 7, line 152, it is stated: “We should note that unlike the configuration using arrayed polarizers [56], this scheme with a polarization-splitting MS does not suffer from a 3-dB intrinsic loss.” This assessment is not supported from inspecting the combined photocurrent in all four channels in the SVR device ($\sim 24 \mu\text{A}$, per Fig. 4c) versus the CR device ($\sim 12 \mu\text{A}$, per Fig. 5c). The photocurrent suggests that practically, there is indeed a -3 dB loss.

5. Page 8, line 174: “Using the fabricated chip, we have successfully demonstrated simultaneous detection of 320-Gbit/s four-channel PAM4 signals by directly attaching a four-core MCF to the chip without using a fan-out device.” This is not true, as the multicore fiber is by previous accounts suspended above and normal to the metasurface, separated by a significant air gap.

6. On line 433, the authors refer to the design of a square lattice nanopost array without further supporting information. The authors should show in the supplementary information the spatial distributions of the meta-atom parameters and conversion tables from D_u and D_v for meta-atoms constructed on the square lattice.

Version 1:

Reviewer comments:

Reviewer #1

(Remarks to the Author)

This study demonstrates a highly innovative optical receiver platform that monolithically integrates functional dielectric metasurfaces with ultrafast membrane photodetector arrays. The work successfully addresses fundamental scalability limitations of conventional photonic integrated circuits by enabling direct surface-normal coupling and massively parallel detection of multi-channel optical signals. The authors have comprehensively responded to all reviewer concerns through rigorous FDTD simulations validating metasurface performance (85% focusing efficiency, >20 dB extinction ratios), experimental quantification of inter-channel crosstalk (<-25 dB), and detailed benchmarking against conventional waveguide-based systems. The demonstrated capabilities—including simultaneous detection of 320 Gbit/s PAM4 signals and coherent detection of 240 Gbit/s 64QAM signals—coupled with the platform's inherent 2D scalability present a transformative approach to optical communications. While some practical aspects like environmental robustness require further study, the manuscript provides sufficient evidence of the technology's potential impact across optical interconnects, quantum communications, and imaging systems. The work represents a significant advance in integrated photonics and meets the high publication standards of Nature Communications.

Reviewer #2

(Remarks to the Author)

The manuscript has improved considerably. The work presents solid contributions to the field, and the revisions have strengthened both the technical content and presentation. I recommend acceptance pending addressing the comments noted below.

Major Comments:

1. Please double-check that reference R15 is indeed the reference with 6 photodetectors, as opposed to R14, and ensure all specifications and references in your comparison table are accurate.

Minor Comments:

2-6. The revisions to the remaining points from the first review are well-executed and address the initial concerns effectively.

We sincerely appreciate the reviewers' insightful comments and suggestions, which have significantly contributed to improving the quality of our paper. Below, we present our response to the individual comments and describe the changes made in the revised manuscript.

Reviewer #1

This manuscript presents a novel optical receiver platform integrating a functional dielectric metasurface with an ultrafast membrane photodetector array, enabling high-speed, parallel detection of multi-channel and multi-modulation-format optical signals. The work demonstrates significant advancements in addressing scalability limitations of traditional photonic integrated circuits and holds promise for applications in space-division multiplexing and coherent optical communications. Below are the areas for improvement.

1. The metasurface design relies heavily on empirical phase and polarization models without rigorous electromagnetic verification. Finite-difference time-domain (FDTD) or rigorous coupled-wave analysis (RCWA) simulations are absent, leaving uncertainties regarding phase uniformity, cross-polarization leakage, and near-field coupling effects. These omissions weaken confidence in the design's reproducibility and scalability.

Response: We thank the reviewer for pointing out an important issue. To investigate the transmission properties of the actual metasurfaces that we designed, we have performed rigorous full-wave simulations using the finite-difference time-domain (FDTD) method. These rigorous simulations, considering near-field coupling between adjacent meta-atoms and reflections at the interfaces, assess the realistic performance of our metasurface design quantitatively. Figures R1–R3 show the simulation results for the single-channel receiver, Stokes-vector receiver (SVR), and coherent receiver (CR), respectively. We can confirm that high focusing efficiencies of 85%, 71%, and 87% are obtained for the single-channel metalens, SVR, and CR, respectively. The desired focusing and polarization functionalities with high efficiency can be clearly confirmed.

Modifications in the revised manuscript:

- In the Supplementary Information, we added Supplement Note 1 with Figs. R1-R3.
- In Line 74, we added a sentence describing FDTD simulation results for the metalens: “*Rigorous full-wave simulation based on a finite-difference time-domain (FDTD) method shows that the designed ML exhibits the desired focusing function with a high focusing efficiency of 85% (see Supplementary Note 1 and Supplementary Fig. 4).*”
- In Line 125, we added a sentence describing SVR simulation results: “*Full-wave simulation with the designed MS shows the desired focusing and polarization-sorting functions with a total focusing efficiency of 71% (see Supplementary Note 1 and Supplementary Fig. 8).*”
- In Line 162, we added a sentence describing CR simulation results: “*Full-wave simulation with the designed MS shows a high focusing efficiency of 87% and low crosstalk below -20 dB (see Supplementary Note 1 and Supplementary Fig. 11).*”

Fig. R1. Full-wave simulation of the metalens. **a**, Simulation model. **b**, Simulated complex electric field distribution at the output of the metalens at 1550-nm wavelength. The dashed lines indicate the metalens aperture. **c**, Electromagnetic power distribution (S_z) at the PD plane. The dashed line indicates the 6- μm PD aperture. **d**, Calculated focusing efficiency as a function of the wavelength.

Fig. R2. Full-wave FDTD simulation of the MS for SVR at 1550-nm wavelength. **a**, Simulated intensity distributions at the PD plane for input polarization states P_1 – P_4 (top) and their orthogonal states P'_1 – P'_4 (bottom). The dashed lines indicate the PD apertures. **b**, Simulated focusing efficiency for each PD and input polarization state.

Fig. R3. Full-wave FDTD simulation of the MS for CR. **a**, Simulated intensity distributions at the PD plane at 1550-nm wavelength for the input light with $\pm 45^\circ$ linear (*a/b*), right/left-handed circular (RHC/LHC; *r/l*) polarization states. **b**, Calculated focusing efficiencies at the four PD ports as a function of wavelength.

2. Critical data on environmental robustness—such as thermal cycling, humidity resistance, and long-term operation—are missing. Such information is vital for assessing practical viability, particularly in harsh telecom environments.

Response: We agree that environmental robustness is essential for practical deployment. Due to limitations in our current laboratory infrastructure, we were unable to experimentally evaluate long-term stability or environmental durability (e.g., under thermal cycling or humidity) in this study. However, we would like to note the following considerations supporting the potential robustness of the proposed device:

- **Photodetector:** The photodetector in our device relies on a hybrid integration of III–V materials on SiO₂, a platform that has been widely studied and established for lasers [R1], modulators [R2], and photodetectors [R3]. In addition, we can employ Ge PDs as an alternative, which are fully CMOS-compatible and already commercialized, providing robust performance and reliability that have been proven in practical applications.
- **Metasurface:** The metasurface itself is a passive, thin optical element with no active moving parts or electrical connections. It is thus stable against the environment under proper conditions. While our proof-of-concept device did not employ a capping layer on the metasurface, incorporating one would further enhance robustness. Moreover, dielectric metasurfaces have already been demonstrated in commercial applications [R4], underscoring their potential environmental stability.

Indeed, comprehensive environmental testing will be crucial for commercialization. However, the main scope of this work is to propose and experimentally validate a new receiver platform, which, we believe, would provide a strong foundation for future development.

Modifications in the revised manuscript:

- In Line 53, we revised the sentence: “*The one-chip receivers were fabricated by first transferring an InGaAs/InP membrane PD layer onto a SiO₂ substrate through the wafer-bonding-based technique that has been extensively developed for integrating III-V active devices with silicon photonics [54, 55].*”
- In Line 404, we added a sentence: “*In future work, a capping layer can be employed on the MSs to enhance device stability and environmental robustness.*”

[R1] C. Xiang, W. Jin, O. Terra, B. Dong, H. Wang, L. Wu, J. Guo, T. J. Morin, E. Hughes, J. Peters, Q.-X. Ji, A. Feshali, M. Paniccia, K. J. Vahala, and J. E. Bowers, “3D integration enables ultralow-noise isolator-free lasers in silicon photonics,” *Nature* **620**, 78–85 (2023).

[R2] J.-H. Han, F. Boeuf, J. Fujikata, S. Takahashi, S. Takagi, and M. Takenaka, “Efficient low-loss InGaAsP/Si hybrid MOS optical modulator,” *Nat. Photonics* **11**, 486–490 (2017).

[R3] K. Nozaki, S. Matsuo, T. Fujii, K. Takeda, M. Ono, A. Shakoor, E. Kuramochi, and M. Notomi, “Photonic-crystal nano-photodetector with ultrasmall capacitance for on-chip light-to-voltage conversion without an amplifier,” *Optica* **3**, 483–492 (2016).

[R4] <https://www.yolegroup.com/technology-outlook/metasurfaces-break-through-turning-speculation-into-reality>

3. The multi-channel receiver’s performance metrics lack quantification of adjacent channel interference (e.g., adjacent channel isolation ratio, ICR). Without this data, it is challenging to evaluate the metasurface’s effectiveness in suppressing inter-channel crosstalk, a key determinant of system-level bit-error-rate (BER).

Response: Following the suggestion, we added new experimental results to evaluate the inter-channel crosstalk. Figure R4 shows the IV curves of the PD in channel 1 under three different conditions: (1) all four cores of the multi-core fiber (MCF) illuminated (red), (2) only non-corresponding ports illuminated (blue), and (3) no illumination (black, dark current). We can confirm that the photocurrent contribution from adjacent channels is suppressed, with residual fluctuations attributable to the measurement noise. From these measurements, we estimate the adjacent channel isolation to be better than -25 dB, which indicates sufficiently low crosstalk and negligible impact on system-level BER under typical operating conditions.

Modifications in the revised manuscript:

- In the Supplementary Information, we added Supplement Note 2, including Fig. R4.
- In Line 110, we added a sentence and reference to Supplementary Note 2: “*crosstalk from other channels is suppressed below -25 dB (see Supplementary Note 2 and Supplementary Fig. 6).*”

Fig. R4. Characterization of crosstalk in the multi-channel receiver. IV curves of the PD at channel 1 measured under three illumination conditions: (1) all channels on, (2) only adjacent channels on, and (3) dark current.

4. The manuscript fails to contextualize the proposed design against established PIC-based receivers (e.g., silicon photonic coherent engines). Direct comparisons of key metrics—power consumption, footprint, latency, or cost—would clarify the innovation’s advantages and limitations.

Response: We appreciate the suggestion to contextualize our design in relation to established silicon photonic integrated circuit (PIC)-based receivers. To address these issues, we added a supplementary discussion section (Supplementary Note 3) that qualitatively compares key metrics, including power consumption, footprint, latency, and cost:

- **Power Consumption:** Since our receiver performs equivalent optical functionality, the same digital signal processing (DSP) can be employed, resulting in comparable DSP-related power consumption to that of existing PIC-based receivers. On the other hand, the surface-normal configuration of our receiver is advantageous in minimizing optical coupling loss, which is critical for enhancing receiver sensitivity and minimizing the energy per bit.
- **Footprint:** Table R1 compares our platform with silicon photonics in terms of footprints. Since our platform does not require edge couplers with spot-size converters (SSCs), which typically range in

length from a few hundred micrometers to a few millimeters [R5, R6], the total footprint can be smaller. In addition, as demonstrated in the four-channel IM-DD receiver (Fig. 3 in the main text) and Ref. [R7], our platform has a superior spatial scalability due to its surface-normal configuration, enabling dense 2D integration of PD arrays and direct coupling from multi-core fibers without SSCs and external bulky fan-out devices.

- **Latency:** Optical-to-electrical conversion and DSP remain unchanged, so latency is comparable to PIC-based approaches.
- **Cost:** Although we have employed in-house fabrication facilities to fabricate InGaAs-membrane devices, such heterogeneous integration platforms have been undergoing rapid commoditization and are in the process of being adopted in silicon photonic processes [R8]. Additionally, high-speed Ge PDs can also be used to enable full CMOS compatibility. With these technologies, we believe our devices can be fabricated at a comparable or lower cost than conventional PIC-based receivers due to their significantly smaller device footprint.

Modifications in the revised manuscript:

- In the Supplementary Information, we added Supplementary Note 3.
- In Line 181, we added a reference to Supplementary Note 3: “(see discussion on the comparison between these platforms in Supplementary Note 3)”

Table R1. Comparison of footprint. MMI: multi-mode interferometer, PBS: polarization beam splitter.

	This work	Si photonics	
	Lateral size (MS diameter)	Lateral size	Components
Single-channel IM-DD receiver	340 μm	$\sim 400 \mu\text{m}$	SSC + PD
Four-channel IM-DD receiver	260 μm	$\sim 500 \mu\text{m}$	4 x SSC + 4 x PD (+ External fanout)
SVR	260 μm	$\sim 1 \text{ mm}$ [R9]	SSC + PBS + BS + 2x4 MMI + 6 x PD
Coherent receiver (single polarization)	120 μm	$\sim 1 \text{ mm}$ [R10]	SSC + 2x4 MMI + 4 x PD

[R5] <https://www.imec-int.com/en/articles/interfacing-silicon-photonics-high-density-co-packaged-optics>

[R6] X. Mu, S. Wu, L. Cheng, and H. Y. Fu, “Edge couplers in silicon photonic integrated circuits: A review,” *Appl. Sci. (Basel)* **10**, 1538 (2020).

[R7] K. Komatsu, G. Soma, S. Ishimura, H. Takahashi, T. Tsuritani, M. Suzuki, Y. Nakano, and T. Tanemura, “Scalable multi-core dual-polarization coherent receiver using a metasurface optical hybrid,” *J. Lightwave Technol.* **42**, 4013–4022 (2024).

[R8] Tower Semiconductor, “OpenLight and Tower Semiconductor Demonstrate 400G/lane Modulators Built on Silicon Photonic Wafers for Data Centers and AI Optical Connectivity,” <https://towersemi.com/2025/03/12/03122025/>

[R9] P. Dong, X. Liu, S. Chandrasekhar, L. L. Buhl, R. Aroca, and Y.-K. Chen, “Monolithic silicon photonic integrated circuits for compact 100 +Gb/s coherent optical receivers and transmitters,” *IEEE J. Sel. Top. Quantum Electron.* **20**, 150–157 (2014).

[R10] P. Dong, X. Chen, K. Kim, S. Chandrasekhar, Y.-K. Chen, and J. H. Sinsky, “128-Gb/s 100-km transmission with direct detection using silicon photonic Stokes vector receiver and I/Q modulator,” *Opt. Express* **24**, 14208–14214 (2016).

5. Experiments are confined to static laboratory conditions (fixed wavelength, polarization, and input power). Real-world applications demand resilience to dynamic perturbations (e.g., polarization drift, wavelength hopping, or fiber nonlinearity). Supplementary tests or discussions on digital signal processing (DSP)-enabled calibration would enhance practical relevance.

Response: Since our metasurface-based coherent receiver is functionally equivalent to standard PIC-based coherent receivers, we can employ the same standard DSP to retrieve the IQ signals. Therefore, dynamic perturbations, such as polarization and wavelength drift, can be automatically calibrated through the DSP-based calibration and compensation techniques.

Modifications in the revised manuscript:

- In Line 501, we added a sentence: “Please note that since the optical hybrid function is identical to that in the standard coherent receiver, we can apply the same DSP to retrieve the IQ signals.”

6. Key figures (e.g., Fig. 2c) lack error bars/confidence intervals, raising concerns about data reliability. Additionally, incomplete definitions of acronyms (e.g., PD, MS) and sparse methodological details hinder reproducibility.

Response: We apologize for the unclear presentation in Fig. 2c. We revised the plot to improve clarity, as shown in Fig. R5. The oscillations observed in the photocurrent spectrum are due to the Fabry-Perot resonance between the output facet of the SMF and the SiO₂-InP interface at the membrane PD. We should note that these variations are not stochastic noise but rather deterministic spectral oscillations obtained under stable, controlled laboratory conditions. Since these data were taken with a sufficiently long integration time of the picoammeter, the error bars are negligible.

Additionally, we have revised figure captions to explicitly define acronyms such as PD and MS, and we have ensured that definitions of all acronyms are included in the main text. Detailed explanations of each measurement setup are provided in the inset of Fig. 2e, the Methods section, and Supplementary Fig. 6.

Modifications in the revised manuscript:

- In Fig. 2, we revised the figure quality.
- In the figure captions, we added the explicit definitions of acronyms.

Fig. R5. Updated Fig. 2c (Photocurrent spectrum).

This work pioneers a metasurface-integrated optical receiver platform with transformative potential for

high-speed, parallel optical communications. However, the absence of rigorous theoretical validation, environmental stability data, and comparative benchmarks limits its impact. Addressing these gaps—through supplementary simulations, and expanded experimental comparisons—would elevate the manuscript to a publishable standard in Nature Communications.

Response: We sincerely thank the reviewer for recognizing the novelty and potential impact of our work. Reviewer's constructive feedback has enabled us to substantially strengthen the manuscript through additional simulations, experiments, and contextual discussion.

Reviewer #2

In this work, the authors present a proof of concept for several integrated metasurface enabled optical receivers for an assortment of compact signal multiplexing schemes. The authors design transmission-mode metasurfaces to image the output of optical fibers onto membrane photodetectors polarization-selectively. This manuscript showcases innovative techniques that advance a new avenue for fabricating high bandwidth optical receivers. I think that the manuscript is suitable for publication in Nature Communications, provided some revisions are made.

Response: We sincerely thank the reviewer for the positive evaluation of our work and for recognizing the novelty and potential of our metasurface-integrated optical receiver architecture.

1. The authors' claim is motivated by the alleged advantage of metasurfaces over conventional PICs, specifically in "their waveguide nature inherently constrain[ing] two dimensional spatial scaling to accommodate a large number of optical signals in parallel." The authors should provide an analysis of the on-chip footprint of equivalent photonic integrated circuitry (injected by, say, an edge coupled fiber) for each device and its scaling behavior for high parallelism.

Response: Table R1 compares our platform with typical silicon photonic receivers in terms of footprints. Since our platform does not require edge couplers with spot-size converters (SSCs), which can range from a few hundred micrometers to a few millimeters in length [R11, R12], the total footprint can be smaller. In addition, as demonstrated in the four-channel IM-DD receiver (Fig. 3 in the main text) and Ref. [R13], our platform has a superior spatial scalability due to its surface-normal configuration, enabling dense 2D integration of PD arrays and direct coupling from multi-core fibers without SSCs or external bulky fan-out devices.

Modifications in the revised manuscript:

- In the Supplementary Information, we added the discussion on the footprint and Table R2 in Supplementary Note 3.
- In Line 181, we added a reference to Supplementary Note 3: "(see discussion on the comparison between these platforms in Supplementary Note 3)"

Table R2. Comparison of footprint. MMI: multi-mode interferometer, PBS: polarization beam splitter.

	This work	Si photonics	
	Lateral size (MS diameter)	Lateral size	Components
Single-channel IM-DD receiver	340 μm	$\sim 400 \mu\text{m}$	SSC + PD
Four-channel IM-DD receiver	260 μm	$\sim 500 \mu\text{m}$	4 x SSC + 4 x PD (+ External fanout)
SVR	260 μm	$\sim 1 \text{ mm}$ [R14]	SSC + PBS + BS + 2x4 MMI + 6 x PD
Coherent receiver (single polarization)	120 μm	$\sim 1 \text{ mm}$ [R15]	SSC + 2x4 MMI + 4 x PD

- [R11] <https://www.imec-int.com/en/articles/interfacing-silicon-photonics-high-density-co-packaged-optics>
- [R12] X. Mu, S. Wu, L. Cheng, and H. Y. Fu, “Edge couplers in silicon photonic integrated circuits: A review,” *Appl. Sci. (Basel)* **10**, 1538 (2020).
- [R13] K. Komatsu, G. Soma, S. Ishimura, H. Takahashi, T. Tsuritani, M. Suzuki, Y. Nakano, and T. Tanemura, “Scalable multi-core dual-polarization coherent receiver using a metasurface optical hybrid,” *J. Lightwave Technol.* **42**, 4013–4022 (2024).
- [R14] P. Dong, X. Liu, S. Chandrasekhar, L. L. Buhl, R. Aroca, and Y.-K. Chen, “Monolithic silicon photonic integrated circuits for compact 100 +Gb/s coherent optical receivers and transmitters,” *IEEE J. Sel. Top. Quantum Electron.* **20**, 150–157 (2014).
- [R15] P. Dong, X. Chen, K. Kim, S. Chandrasekhar, Y.-K. Chen, and J. H. Sinsky, “128-Gb/s 100-km transmission with direct detection using silicon photonic Stokes vector receiver and I/Q modulator,” *Opt. Express* **24**, 14208–14214 (2016).

2. Data in Figs. 2c and 2e are plotted somehow with a variable thickness line, which at some points seems as if it is multivalued with respect to the frequency axis. The authors should verify that the data is valid and plot again with conventional data lines and data markers.

Response: We apologize for the unclear plot in Figs. 2c,e. We updated the figures with clear lines to avoid misinterpretation (Fig. R6).

Modifications in the revised manuscript

- We revised Fig. 2c,e.

Fig. R6. Updated Fig. 2c,e.

3. In Figure S3b, the transmitted phase is calculated as a function of the structural parameter D . However, Figure S5b presents the structural parameter as a function of desired $\varphi_u(x, y)$ and $\varphi_v(x, y)$ instead of vice versa. Additionally it is claimed that this mapping is “one-to-one” on line 422 and elsewhere. This is highly unlikely, seeing that the relation shown in S3b is not one-to-one. The authors should show φ_u , φ_v , and T as a function of D_u and D_v , and remove the language “one-to-one.”

Response: We appreciate the reviewer for carefully checking the details of our work, and we apologize for the lack of information. In our workflow, we first simulate the transmission properties as a function of structural parameters (Figs. R7c, R8c, and R9c). Then, we derive the map, which converts the required phases into the structural parameters (Figs. R7d, R8d, and R9d).

For the elliptical nanopost case, for example, we first simulate the transmission $t_u(D_u, D_v)$ and $t_v(D_u, D_v)$ in a periodic array of Si nanoposts using rigorous coupled-wave analysis (RCWA) (Figs. R8c and R9c).

Then, using these results, we derive the geometrical parameters ($D_u(\varphi_u, \varphi_v)$, $D_v(\varphi_u, \varphi_v)$) to satisfy the target phase shifts (φ_u, φ_v) with a high transmittance (Figs. R8d and R9d):

$$(D_u(\varphi_u, \varphi_v), D_v(\varphi_u, \varphi_v)) = \underset{D_u, D_v}{\operatorname{argmin}} \left\{ |\tilde{t}_u(D_u, D_v) - e^{i\varphi_u}|^2 + |\tilde{t}_v(D_u, D_v) - e^{i\varphi_v}|^2 \right\},$$

As the reviewer pointed out, Figs. R7d and R8c are missing from the previous manuscript. We added them in the revised manuscript for ease of understanding. In addition, we removed the term “one-to-one.”

Modifications in the revised manuscript:

- In the Supplementary Information, we added Figs. R7-9 as Supplement Figs. 3, 7, and 10.
- In Line 437, we expanded a description of parameter derivation.
- In Lines 427, 444, and 456, we removed “one-to-one.”

Fig. R7. ML design for the IM-DD receiver. **a**, Required phase profile $\varphi_{\text{ML}}(x, y)$ to realize the ML of the IM-DD receiver in Fig. 2. **b**, Schematic of a periodic array of circular Si nanoposts arranged on a triangular lattice with a lattice constant of 700 nm and $\theta = 0$. **c**, Simulated transmittance $|\tilde{t}|^2$ and phase $L\tilde{t}/2\pi$ for a periodic Si nanopost array on a triangular lattice with a lattice constant of 700 nm (inset). The shaded region is excluded in the ML design to maintain high transmission. **d**, Required post diameter D to realize the required transmission phase. **e**, Spatial distribution of the designed post diameter $D(x, y)$

Fig. R8. MS design for the SVR. **a**, Spatial distributions of the optical parameters $\varphi_u(x, y)$, $\varphi_v(x, y)$, and $\theta(x, y)$ for the SVR. **b**, Schematic of a periodic array of elliptical Si nanoposts arranged on a triangular lattice with a lattice constant of 700 nm and $\theta = 0$. **c**, Simulated transmission coefficients for x- and y-polarized inputs as functions of meta-atom dimensions (D_u, D_v). **d**, Optimal dimensions (D_u, D_v) of elliptical Si nanoposts that provide phase shifts of (φ_u, φ_v) to the x- and y-polarized transmitted light. The dimensions were selected from 200 to 600 nm for ease of fabrication. **e**, Spatial distributions of the designed post dimensions $D_u(x, y)$ and $D_v(x, y)$.

Fig. R9. MS design for the CR. **a**, Spatial distributions of the optical parameters $\varphi_u(x, y)$, $\varphi_v(x, y)$, and $\theta(x, y)$ for the CR. **b**, Schematic of a periodic array of elliptical Si nanoposts arranged on a square lattice with a lattice constant of 700 nm and $\theta = 0$. **c**, Simulated transmission coefficients for x- and y-polarized inputs as functions of meta-atom dimensions (D_u, D_v). **d**, Optimal dimensions (D_u, D_v) of elliptical Si nanoposts that provide phase shifts of (φ_u, φ_v) to the x- and y-polarized transmitted light. The dimensions were selected from 200 to 600 nm for ease of fabrication. **e**, Spatial distributions of the designed post dimensions $D_u(x, y)$ and $D_v(x, y)$.

4. On page 7, line 152, it is stated: “We should note that unlike the configuration using arrayed polarizers [56], this scheme with a polarization-splitting MS does not suffer from a 3-dB intrinsic loss.” This assessment is not supported from inspecting the combined photocurrent in all four channels in the SVR device ($\sim 24 \mu\text{A}$, per Fig. 4c) versus the CR device ($\sim 12 \mu\text{A}$, per Fig. 5c). The photocurrent suggests that practically, there is indeed a -3 dB loss.

Response: We appreciate the reviewer for pointing out this inconsistency. It is indeed true that our fabricated CR device somehow exhibited smaller responsivity ($\sim 0.13 \text{ A/W}$) compared with the SVR device ($\sim 0.22 \text{ A/W}$). Although the exact reason is not clear at the moment, we attribute this to the larger fabrication imperfection of the CR device since it was located near the chip edge.

Theoretically, however, the ideal metasurface design should not suffer a 3-dB intrinsic loss because the light is split by polarization rather than filtered. To confirm this, we have performed an additional full-wave simulation of the CR device. As shown in Fig. R10, we can confirm that desired polarization sorting with high efficiency (84%, i.e., 0.8 dB loss) can be obtained, suggesting that improved fabrication could exceed the -3 dB limit of the polarizer-based CRs.

Modifications in the revised manuscript:

- In Line 170, we added a sentence: “*The rather small total responsivity ($\sim 0.13 \text{ A/W}$), corresponding to the focusing efficiency of $\sim 40\%$, is attributed to imperfect MS fabrication and alignment error of the input fiber.*”
- In the Supplementary Information, we added the full-wave simulation results in the Supplement Note 1.

Fig. R10. Full-wave FDTD simulation of the MS for CR. **a**, Simulated intensity distributions at the PD plane at 1550-nm wavelength for the input light with $\pm 45^\circ$ linear (*a/b*), right/left-handed circular (RHC/LHC; *r/l*) polarization states. **b**, Calculated focusing efficiencies at the four PD ports as a function of wavelength.

5. Page 8, line 174: “Using the fabricated chip, we have successfully demonstrated simultaneous detection of 320-Gbit/s four-channel PAM4 signals by directly attaching a four-core MCF to the chip without using a fan-out device.” This is not true, as the multicore fiber is by previous accounts suspended above and normal to the metasurface, separated by a significant air gap.

Response: We apologize for the misleading expression. Our intention was to highlight the elimination of bulky fan-out devices, which typically exceed 1 cm in length, while maintaining an air gap of less than 1 mm. To avoid confusion, we removed the expression “by directly attaching.”

Modifications in the revised manuscript

- In Line 186, we revised the sentence: “*Using the fabricated chip, we have successfully demonstrated simultaneous detection of 320-Gbit/s four-channel PAM4 signals from a four-core MCF without using a fan-out device.*”

6. On line 433, the authors refer to the design of a square lattice nanopost array without further supporting information. The authors should show in the supplementary information the spatial distributions of the meta-atom parameters and conversion tables from D_u and D_v for meta-atoms constructed on the square lattice.

Response: We apologize for the lack of information. We added supplementary figures showing the spatial distributions of meta-atom parameters (Figs. R9a,e) and conversion tables from D_u and D_v for square-lattice nanoposts (Figs. R9b-d).

In the design of the CR, we first derive the required phase profiles in MS-A and MS-B to focus the input light onto the desired PD positions. Then, we calculate the required optical parameters $\varphi_u(x, y)$, $\varphi_v(x, y)$, and $\theta(x, y)$ (Fig. R9a). Finally, using the look-up table shown in Fig. R9d, we obtain the ellipse dimensions of each meta-atom: $D_u(x, y)$ and $D_v(x, y)$ (Fig. R9e).

Modifications in the revised manuscript:

- In Supplementary Information, we added Fig. R9 as Supplementary Fig. 10
- In “MS for CR” in the Methods section (Line 448), we updated the sentences with reference to Supplementary Fig. 10.

We would like to thank the Reviewers once again for their time and effort in reviewing our paper. Below, we present our response to the comments from the Reviewers.

Reviewer #1

This study demonstrates a highly innovative optical receiver platform that monolithically integrates functional dielectric metasurfaces with ultrafast membrane photodetector arrays. The work successfully addresses fundamental scalability limitations of conventional photonic integrated circuits by enabling direct surface-normal coupling and massively parallel detection of multi-channel optical signals. The authors have comprehensively responded to all reviewer concerns through rigorous FDTD simulations validating metasurface performance (85% focusing efficiency, >20 dB extinction ratios), experimental quantification of inter-channel crosstalk (<-25 dB), and detailed benchmarking against conventional waveguide-based systems. The demonstrated capabilities—including simultaneous detection of 320 Gbit/s PAM4 signals and coherent detection of 240 Gbit/s 64QAM signals—coupled with the platform's inherent 2D scalability present a transformative approach to optical communications. While some practical aspects like environmental robustness require further study, the manuscript provides sufficient evidence of the technology's potential impact across optical interconnects, quantum communications, and imaging systems. The work represents a significant advance in integrated photonics and meets the high publication standards of Nature Communications.

Reply: We appreciate the reviewer for the positive assessment.

Reviewer #2

The manuscript has improved considerably. The work presents solid contributions to the field, and the revisions have strengthened both the technical content and presentation. I recommend acceptance pending addressing the comments noted below.

Reply: We appreciate the reviewer for the positive assessment.

Major Comments:

1. Please double-check that reference R15 is indeed the reference with 6 photodetectors, as opposed to R14, and ensure all specifications and references in your comparison table are accurate.

Reply: We apologize for the mistake. Indeed, R14 and R15 were inverted in the response letter. It should

be corrected as Table R1. We have confirmed that all reference and specifications listed in Table S1 of the Supplementary Information are correct.

Table R1. Comparison of footprint. MMI: multi-mode interferometer, PBS: polarization beam splitter.

	This work	Si photonics	
	Lateral size (MS diameter)	Lateral size	Components
Single-channel IM-DD receiver	340 μm	$\sim 400 \mu\text{m}$	SSC + PD
Four-channel IM-DD receiver	260 μm	$\sim 500 \mu\text{m}$	4 x SSC + 4 x PD (+ External fanout)
SVR	260 μm	$\sim 1 \text{ mm}$ [R15]	SSC + PBS + BS + 2x4 MMI + 6 x PD
Coherent receiver (single polarization)	120 μm	$\sim 1 \text{ mm}$ [R14]	SSC + 2x4 MMI + 4 x PD

[R14] P. Dong, X. Liu, S. Chandrasekhar, L. L. Buhl, R. Aroca, and Y.-K. Chen, “Monolithic silicon photonic integrated circuits for compact 100 +Gb/s coherent optical receivers and transmitters,” *IEEE J. Sel. Top. Quantum Electron.* 20, 150–157 (2014).

[R15] P. Dong, X. Chen, K. Kim, S. Chandrasekhar, Y.-K. Chen, and J. H. Sinsky, “128-Gb/s 100-km transmission with direct detection using silicon photonic Stokes vector receiver and I/Q modulator,” *Opt. Express* 24, 14208–14214 (2016).

Minor Comments:

2-6. The revisions to the remaining points from the first review are well-executed and address the initial concerns effectively.

Reply: We appreciate the reviewer’s positive assessment. We are glad that our revisions have addressed the initial concerns.